# Natural Product Library Screens Identify Sanguinarine Chloride as a Potent Inhibitor of Telomerase Expression and Activity

**DOI:** 10.3390/cells11091485

**Published:** 2022-04-28

**Authors:** Siyu Yan, Song Lin, Kexin Chen, Shanshan Yin, Haoyue Peng, Nanshuo Cai, Wenbin Ma, Zhou Songyang, Yan Huang

**Affiliations:** 1MOE Key Laboratory of Gene Function and Regulation, Guangzhou Key Laboratory of Healthy Aging Research and State Key Laboratory of Biocontrol, School of Life Sciences, Sun Yat-Sen University, Guangzhou 510275, China; yansy3@mail2.sysu.edu.cn (S.Y.); lins3@mail2.sysu.edu.cn (S.L.); chenkx25@mail2.sysu.edu.cn (K.C.); yinshsh2@mail2.sysu.edu.cn (S.Y.); penghy8@mail2.sysu.edu.cn (H.P.); cainsh@mail2.sysu.edu.cn (N.C.); mawenbin@mail.sysu.edu.cn (W.M.); songyanz@mail.sysu.edu.cn (Z.S.); 2Sun Yat-Sen Memorial Hospital, Sun Yat-Sen University, Guangzhou 510120, China; 3Bioland Laboratory (Guangzhou Regenerative Medicine and Health Guangdong Laboratory), Guangzhou 510005, China

**Keywords:** telomerase/hTERT, anti-cancer, sanguinarine chloride, cellular senescence

## Abstract

Reverse transcriptase hTERT is essential to telomerase function in stem cells, as well as in 85–90% of human cancers. Its high expression in stem cells or cancer cells has made telomerase/hTERT an attractive therapeutic target for anti-aging and anti-tumor applications. In this study, we screened a natural product library containing 800 compounds using an endogenous hTERT reporter. Eight candidates have been identified, in which sanguinarine chloride (SC) and brazilin (Braz) were selected due to their leading inhibition. SC could induce an acute and strong suppressive effect on the expression of *hTERT* and telomerase activity in multiple cancer cells, whereas Braz selectively inhibited telomerase in certain types of cancer cells. Remarkably, SC long-term treatment could cause telomere attrition and cell growth retardation, which lead to senescence features in cancer cells, such as the accumulation of senescence-associated β-galactosidase (SA-β-gal)-positive cells, the upregulation of p16/p21/p53 pathways and telomere dysfunction-induced foci (TIFs). Additionally, SC exhibited excellent capabilities of anti-tumorigenesis, both in vitro and in vivo. In the mechanism, the compound down-regulated several active transcription factors including p65, a subunit of NF-κB complex, and reintroducing p65 could alleviate its suppression of the hTERT/telomerase. Moreover, SC could directly bind hTERT and inhibit telomerase activity in vitro. In conclusion, we identified that SC not only down-regulates the *hTERT* gene’s expression, but also directly affects telomerase/hTERT. The dual function makes this compound an attractive drug candidate for anti-tumor therapy.

## 1. Introduction

Telomeres are continuously shortened during the process of DNA replication as DNA polymerase cannot synthesize chromosomal end sequences [1]. Mammalian telomeric DNA consists of TTAGGG hexanucleotide tandem DNA repeats forming loop structures, with the interactions of specialized telosome/shelterin proteins [2,3]. In order to maintain the genomic stability and protect cells from senescence, shortened telomeres could be elongated by two mechanisms: in a telomerase-dependent manner in most pluripotent stem cells and cancer cells, or the alternative lengthening of telomeres (ALT) in 10–15% cancer cells [4].

Human telomerase ribonucleoprotein contains the catalytic reverse transcriptase hTERT and RNA template hTERC associated with the accessory H/ACA proteins [5]. hTERT is highly conserved among species. Human TERT is composed of four main domains [6]. The N-terminal extension (TEN) domain is connected with the 5′-terminal of the telomerase RNA-binding domain (TRBD) by a short linker sequence. The central reverse transcriptase (RT) domain and the C-terminal extension (CTE) domain make up a right-hand structure containing palm-, finger- and thumb-like subdomains. Together, the sequential TRBD-RT-CTE domains form a ring-like structure for telomere repeat addition [6].

Telomerase activity is strictly modulated at multiple levels, including the expression of telomerase subunits, the process of holoenzyme assembly and its recruitment to telomeres [7]. Compared with the ubiquitously expressed hTERC in cells, the restrictively expressed protein hTERT is the major limiting factor of telomerase activity regulation [8]. Suppressing telomerase activity by decreasing hTERT protein blocks telomere extension. Very short telomeres could trigger telomere dysfunction and induce a DNA damage response and cell senescence. Except for its involvement in telomeres elongation, hTERT turned out to participate in many non-telomeric biological events; for instance, the expressional regulation of aging-related genes or oncogenes [9]. Interestingly, c-myc can bind to the E-box motif in the *hTERT* promoter region to activate its transcription [10], while hTERT is also able to stabilize c-myc protein and modulates its binding to target promoters [11]. NF-κB p65 has been reported to modulate telomerase expression [12], and also mediate the nuclear translocation of the hTERT protein from cytoplasm via TNF-α in the cancer cell line [13].

In most cancer cells, stem/progenitor cells and certain somatic cells in special physiological states, such as activated T cells, hTERT is expressed to activate telomerase, making it attractive as a therapeutic target for cancer [14,15]. Imetelstat, also known as GRN163L, is chemically modified oligonucleotide, which can silence the telomerase assembly process [16]. BIBR1532 has been identified as a selective telomerase inhibitor that tightly binds to the FVYL motif near TRBD and can result in an impediment to the interaction of hTERT TRBD with the CR4/5 stem loop of telomerase RNA [17]. Yet, like most quinoline derivatives, BIBR1532 exhibits a certain degree of cytotoxicity, and causes apoptosis and senescence in high doses over 25 μM [18].

Cell senescence is a state of growth arrest caused by several factors, such as telomere loss and DNA damage [19]. Cancer cells show features such as senescence after exposure to certain chemotherapeutic compounds [20]. Therefore, the telomerase inhibitor, as a factor of accelerated cell senescence, is a double-edged sword on its applications, and is accepted in anti-cancer strategies via triggering senescence.

Successful clinical outcomes require prolonged treatment, which may lead to severe toxicity in patients. In comparison with synthetic chemicals, natural products are more acceptable and environmentally friendly. In this work, we used CRISPR/Cas9 to establish a cell-based platform aiming to screen out certain natural compounds that modulate the expression of endogenous *hTERT*, and found some compounds with potential applications, of which sanguinarine chloride (SC), a benzophenanthridine alkaloid extracted from the root of Sanguinaria canadensis, is very attractive.

SC exhibits clear-cut antitumor properties, with evidence of apoptotic cell death induction and anti-proliferation through generating reactive oxygen species [21], suppressing the NF-κB pathway [22], inhibiting cyclin-dependent kinases and cyclins [23] and blocking VEGF function in angiogenesis [24]. Besides this, sanguinarine is commonly used in toothpaste and oral health products because of its antibacterial and anti-inflammatory effects [25]. However, since 1999, a sanguinarine-added mouthwash product Viadent^®^ was reported to be associated with age-related leukoplakia, indicating its pre-neoplastic adverse effects [26,27]. Notably, how sanguinarine induces leukoplakia remains unclear so far, and the underlying mechanism of the anti-tumor effect of sanguinarine remains elusive; thus, figuring out its biological target and detailed molecular mechanism is crucial for pharmacological usage.

## 2. Materials and Methods

### 2.1. Chemicals

Sanguinarine chloride hydrate (SC) was purchased from Aladdin (S101540; Shanghai, China) and Brazilin (Braz) was purchased from Sigma-Aldrich (SML2132; St. Louis, MO, USA). All the chemicals were dissolved in DMSO and stored at −20 °C.

### 2.2. Cell Culture and Transfection

The HEK293T cell line and cancer cell lines, including HTC75, HeLa, DLD1, MDA-MB-231, Hs578t and A549, were routinely cultured in Dulbecco’s Modified Eagle’s Medium (DMEM; Corning; New York, NY, USA) supplemented with 10% Fetal bovine serum (FBS; Excell Bio; Jiangsu, China). Human skin fibroblasts (HFs) and human umbilical vein smooth muscle cells (HUVSMCs) were cultured in Dulbecco’s Modified Eagle Medium/F-12 Nutrition Mixture (DMEM/F12; Gibco; New York, NY, USA) containing 10% FBS (Hyclone; Logan, UT, USA). Peripheral blood mononuclear cells (PBMCs) were cultured in RPMI 1640 medium (Gibco; New York, NY, USA) with 10% FBS (Hyclone; Logan, UT, USA). Lipofectamine2000 reagents (Invitrogen; Carlsbad, CA, USA) were used for cell transfections of recombinant plasmids.

### 2.3. Flow Cytometry Screening

A total of 800 compounds of a natural product library (Natural Products Collection; Microsource; Gaylordsville, CT, USA) was applied and screened in a hTERT-P2A-GFP reporter cell line. Cells were treated by compound in 96wells for 48 h and were then harvested in PBS buffer. GFP and dsRed2 expressions were analyzed by using flow cytometry (Beckman CytoFLEX S; Brea, CA, USA). The mean fluorescence intensity (MFI) of collected cells was taken as the screening index indicating the expression of the target gene.

### 2.4. Quantitative Telomeric Repeat Amplification Protocol (Q-TRAP) and IP-TRAP

Q-TRAP assays were performed as described [28]. Briefly, 10^5^ cells were lysed on ice for 30 min in 100 μL NP40 lysis buffer (10 mM Tris-HCl pH 8.0; 1 mM MgCl_2_; 1 mM EDTA; 0.25 mM sodium deoxycholate; 150 mM NaCl; 1% NP-40; 10% glycerol; 1% fresh protease inhibitor cocktail) and then centrifuged at 13,200 rpm for 10 min at 4 °C. The supernatant was mixed with 100 ng/μL TS primer (5′-AATCCGTCGAGCAGAGTT-3′), 100 ng/μL ACX primer (5′-GCGCGGCTTACCCTTACCCTTACCCTAACC-3′), and 1mM EGTA in 2 × RealStar Green Power Mixture (with ROX) (GeneStar; Beijing, China), and then incubated at 30 °C for 30 min for telomeric repeat extension and PCR amplification (40 cycles, 95 °C for 15 s and 60 °C for 60 s) using the Step One Plus^TM^ Real-Time PCR system (Applied Biosystems; Foster, CA, USA). As for IP-TRAP, the cell lysis was immunoprecipitated by anti-FLAG M2 beads (Sigma-Aldrich; St. Louis, MO, USA) at 4 °C for 3 h and eluted by 3 × FLAG peptides. The eluates were mixed with the compound and subjected to the TRAP assay.

### 2.5. Terminal Restriction Fragment (TRF)

The average telomere length was measured as described [29]. Briefly, genomic DNA was digested by Hinf I and Rsa I overnight at 37 °C, separated by agarose gel, then denatured and hybridized with a radio labeled telomeric probe (TTAGGG)**_4_**. The dried gel was exposed to a phosphor screen and then scanned with Amersham Typhoon IP Phosphorimager (GE Healthcare; Torrington, CT, USA). The average telomere length was calculated using ImageJ (National Institutes of Health developed; Bethesda, MD, USA) and GraphPad Prism software (San Diego, CA, USA).

### 2.6. SA-β-Gal Staining

The assay was performed by using a Senescence β-Galactosidase Staining Kit (Beyotime; Shanghai, China). Briefly, cells were seeded in 12-well plates and fixed with 4% formaldehyde for 15 min at room temperature. The fixed cells were then washed with PBS 3 times and incubated with fresh SA-β-gal staining reagent mix containing 1.0 mg/mL X-galactosidase at 37 °C for 24 h to microscopically observe the staining.

### 2.7. GST Pull Down and Telomerase Activity Reconstitution In Vitro

The GST pull down assay for GST-hTERT purification was carried out as described previously [30]. Briefly, IPTG was added at 16 °C for 20 h to stimulate the hTERT fusion protein expression. The fusion protein was incubated with GST beads at 4 °C for 4 h and washed three times. The eluates were stored at −80 °C for further experiments.

The reconstitution of telomerase activity in vitro was performed as described [28]. Purified GST-tagged hTERT products (GST-opTERT) were incubated with in vitro transcribed hTERC in telomerase reconstruction buffer (25 mM Tris-HCl pH7.4; 2.6 mM KCl; 1 mM MgCl_2_; 136 mM NaCl; 1 mM EGTA; 10% glycerol; 1 mM DTT; 1×proteinase inhibitor cocktail; 0.5 U/μL of RNase inhibitor) at 37 °C for 30 min. Compounds in serially diluted concentrations were added into the reconstructive products, followed by the TRAP assay.

### 2.8. Thioflavin T (ThT) Biochemical Assay

The experiments were conducted in 96-well microplates. In total, 1 μg genome of DNA sample was mixed with ThT at a final concentration of 2 μM in the buffer (20 mM Tris-HCl pH 7.0, 40 mM KCl) at room temperature. The fluorescence emission was collected at 491 nm in a multi-mode microplate reader (BioTek; Winooski, VT, USA).

### 2.9. Microscale Thermophoresis (MST) Assay

Human telomeric oligonucleotides (Telo24, 5′-Cy5-(TTAGGG)_4_-3′) were annealed in the K+ buffer (10 mM K_2_HPO_4_/KH_2_PO_4_ pH 7.0, 100 mM KCl) by heating to 95 °C for 6 min, then cooled down to room temperature and store at 4 °C. The annealed telomeric G-quadruplex samples (1 μM) were incubated with compound SC at concentrations ranging from 0.15625 μM to 320 μM in the K^+^ buffer for 30 min, followed by the MST assays. The MST assay was conducted using the Monolith NT.115 device (NanoTemper Technologies; Munich, Germany) according to the manufacturer’s instructions. Data were analyzed using MO. Affinity Analysis software (NanoTemper Technologies; Munich, Germany).

### 2.10. Fluorescence Polarization Assay

SC at a high concentration over 10 μM exhibits autofluorescence. The equilibrium binding of the compound with hTERT TRBD protein was monitored by fluorescence polarization assay. All fluorescence polarization compound–protein binding assays were performed in 100 μL PBS buffer containing 10 μM SC and purified His tagged hTRBD in a serially diluted concentration from 2.5 μM to 40 μM in 96-well black polypropylene plates. Fluorescence polarization (FP) measurements were performed at room temperature using a Victor^TM^ X5 2030 Multiple Reader (PerkinElmer; Waltham, MA, USA). BSA was used as a negative control.

### 2.11. Soft-Agar Colony Formation Assay

The MDA-MB-231 cell suspension was mixed in 0.3% soft agar in DMEM containing 10% FBS and the compound, then layered on 0.6% solid agar in DMEM containing 10% FBS and the compound. In total, 1000 cells were seeded per well in a 6-well plate. After 14 days of culturing, colonies were observed under a microscope and the total numbers of colonies from ten random fields of view were counted for the statistical analysis.

### 2.12. In Vivo Cell Derived Xenograft (CDX)

MDA-MB-231 cells suspended in cold PBS buffer were inoculated subcutaneously into 6–8-week-old nude mice in situ. After the xenograft model was established, the mice were injected intravenously with the compound at a dosage of 1.1 μg/kg (the concentration of the compound in blood was 1 μM if the blood volume was estimated as 7% of the body weight). The compound was administrated every three days. The volume of tumor and the body weight were recorded before each injection. The tumor volume was calculated as LW^2^/2, where L represents the long diameter and W represents the short diameter.

### 2.13. Statistical Analysis

Data are shown as mean ± SD. Experiments were carried out in three technical replicates. Student’s t-test and one-way ANOVA test were used for statistical significance analyses with the software GraphPad Prism version 6.0 (San Diego, CA, USA). The fitting curves were depicted using Original version 9.0. A *p* value less than 0.05 is statistically significant (* *p* < 0.05, ** *p* < 0.01, *** *p* < 0.001, **** *p* < 0.0001).

## 3. Results

### 3.1. Natural Compound Screening in hTERT Promoter-Driven GFP Reporter Cell Line and Identification of Potential Inhibitor Candidates

In order to screen out *hTERT* regulatory molecules, we established a platform of *hTERT* reporter HEK293T cells using CRISPR/Cas9 to knock in the P2A-GFP fusion gene before the stop codon of the *hTERT* gene (Figure 1A). *hTERT* and *GFP* were transcribed together from the same promoter, and translated fusion proteins were self-cleaved by a small linker peptide, P2A. The red fluorescent protein dsRed2 was stably transfected into the reporter cell line as an internal control of the fluorescence-based FACs screening. Since the abundance of the hTERT protein was much lower than that of most of the other proteins in the cell, here, we took advantage of a monoclonal-derived cell line with a relatively low intensity of GFP and a high intensity of dsRed2 for the screening (Figure 1A and Appendix A). The initial screening focused on the commercial natural compound library containing 800 small molecules (Appendix A); 69 compounds exhibiting the mean fluorescence intensity (MFI) of GFP, normalized by dsRed2 with at least 40% decline compared to the control (DMSO), were enriched for the second screening (Appendix A). These 69 natural products were conducted to three independent repetitive screens and 8 candidate compounds were repeatably obtained with a significant decrease in the MFI of GFP/dsRed2 (Figure 1B, Appendix A). These eight candidates were further verified, and SC and Braz were selected due to their outstanding inhibitory effects (Figure 1C and Appendix A). Braz has been patented as a kind of natural telomerase inhibitor [31].

Based on the flow cytometry data, treating reporter cells with SC (1 μM) or Braz (10 μM) for 48 h decreased the MFI of GFP, but did not influence dsRed2 expression (Figure 1D and Appendix A). The *hTERT* expression level and relative telomerase activity (RTA) were examined in reporter cells under treatment with SC or Braz. Indeed, both SC and Braz inhibited *hTERT* mRNA level and telomerase activity, which confirms our screening result (Figure 1E,F and Appendix A).

### 3.2. Inhibitory Effects of SC on Telomerase Activity in HTC75 Cancer Cells

The characteristic of its higher expression in most cancer cells makes telomerase/hTERT a valuable predictive biomarker and drug target in malignant cells. To evaluate their inhibition of RTA in cancer cells, we treated HTC75 cells with the two candidate compounds respectively for 48 h, and then performed the Q-TRAP assay. HTC75 is a telomerase-positive fibrosarcoma cell line that can maintain a constant telomere length during in vitro passaging, and is commonly used in the telomere field [32]. The results suggested that SC suppresses the telomerase activity in a dose-dependent manner (Figure 2A). The CCK-8 cell proliferation assay showed the effect of SC at different dosages on the viability of HTC75 cell. The fitting curve indicated the median viable concentration was 2.18 μM (Figure 2B). To explore cancer cell proliferative inhibition induced by the compound, we carried out an analysis of cell cycle and apoptosis. SC-treated cells showed a subtle cell cycle arrest in the G2/M phase compared with control cells (Figure 2C,D). Cells incubated with 2 μM SC exhibited an acute increase in apoptotic cells (Figure 2E). The result was consistent with previously reported studies, wherein SC induced apoptosis through generating reactive oxygen species [33,34]. Furthermore, we wondered if SC effectively works in different types of cancer cells; the RTA of five other kinds of solid tumor cell lines were examined after 48 h treatment. SC exhibited a consistent suppressive effect on telomerase modulation in all tested cell lines, although with different degrees of inhibitory effects (Figure 2F).

Compared with SC, Braz could also induce cell cycle arrest in the G2/M phase (Appendix A), whereas no exacerbation of apoptosis events was observed at the concentrations that inhibit telomerase activities in HTC75 cells (Appendix A). In terms of the potential use as a natural telomerase inhibitor, Braz could only inhibit telomerase in certain types of cancer cells, which indicates its restricted applicability in multiple kinds of tumors, compared to the broad-spectrum anti-telomerase property of the compound SC (Appendix A). Furthermore, we assessed the cytotoxicity of Braz in different cancer cells. The compound showed proliferative inhibition effects in HeLa, DLD1 and HTC75 cells, which also inhibit RTA in these three cell lines (Appendix A).

### 3.3. Effects of SC on Cancer Cell Senescence and Telomere Length through Prolonged Treatment

To identify the optimal anti-telomerase dosage of SC, we evaluated RTA in HTC75 tumor cells treated with SC at different concentrations for 48 h. The Q-TRAP results showed the IC50 (half maximal inhibitory concentration) of SC to telomerase was 1.21 μM (Figure 3A). Cell cycle arrest and apoptosis events may indirectly down-regulate the telomerase. These indirect negative cellular events should be avoided whenever possible when the telomerase inhibitor is applied in anti-tumor therapy. Based on these considerations and the data mentioned above, 1 μM SC could substantially inhibit telomerase activity with no obvious apoptosis induction in cancer cells. Therefore, we chose this concentration for further experiments.

Upon continuous treatment with the specified dose of compound SC, the HTC75 cells retained unchanged morphological characteristics compared to the DMSO-treated control cells in a short period; however, prolonged SC-treated cells became shrunken and irregular (Figure 3B). A curve of cumulative population doubling was plotted to assess cancer cell growth. The proliferation of SC-treated cells was much slower than the DMSO-treated cells (Figure 3C). The persistent inhibition of hTERT protein level caused by prolonged compound treatment (Figure 3D) may cause the accumulation of telomere attrition. Telomere length was measured by the terminal restriction fragment (TRF) assay. The average telomere length of HTC75 cells treated with SC for a long time was obviously shortened, compared to the DMSO-treated HTC75 cells (Figure 3E,F). The shortened telomeres can manifest a DNA damage response and drive the cells into senescence. SA-β-gal staining showed more senescent cells in the SC-treated group (Figure 3G). The senescent markers p16/p21/p53 were all up-regulated at the protein levels (Figure 3H). Prolonged SC treatment also induced more telomere dysfunction-induced foci (TIFs) in HTC75 cells (Figure 3I).

As a moderate telomerase inhibitor in cancer cells, Braz suppressed HTC75 cell growth as well as RTA in a long-period treatment (Appendix A). However, the average telomere length of Braz-treated cells remained unchanged compared to that of the control cells (Appendix A).

### 3.4. SC Inhibits Telomerase Depending on p65 Expression

We have confirmed that SC could reduce the mRNA level of *hTERT* gene; the dual luciferase reporter assay of *hTERT* promoter (−1200 bp) suggested the suppressive effect of SC on transcriptional activity (Figure 4A). In addition, we also treated the cancer cells with the transcription blocking reagent Actinomycin D and SC to assess the mRNA stability of *hTERT*, and found that SC does not affect *hTERT* mRNA stability (Appendix A). So far, we have speculated that SC suppresses *hTERT* expression by modulating *hTERT* promoter activity, but not mRNA stability.

The *hTERT* promoter contains many transcription factor-binding sites, including GC-motifs and E-boxes, which can directly modulate telomerase transcription in response to physiological processes, including tumorigenesis. The transcriptional regulation of telomerase is complicated in different cancer cells due to diversified mutations and multi-layered networks [35]. For cells chronically exposed to the compound SC, we detected the mRNA levels of 10 previously reported classic transcription factors and found a significant decrease in *p65*, *c-MYC*, and *MXD1* levels. Among the positively correlated transcription factor genes, *p65* was observed to be down-regulated by SC to the most significant extent (Figure 4B and Appendix A). Moreover, Western blotting confirmed a decreased level of the p65 protein in the cells treated with the compound (Figure 4C).

To investigate the mechanism by which SC inhibits hTERT/telomerase, we transiently transfected p65 and c-myc plasmids respectively into cancer cells treated with the compound (Figure 4D and Appendix A). The reintroduction of p65, rather than c-myc, alleviated the inhibitory effect of SC on telomerase at both the *hTERT* mRNA level (Figure 4E and Appendix A) and the RTA level (Figure 4F and Appendix A). These results indicate that SC inhibits hTERT/telomerase in a p65-dependent manner.

### 3.5. SC Directly Modulates Telomerase In Vitro

The rapid attrition of telomere length in SC-treated cells suggests that SC might also directly inhibit telomerase activity, besides decreasing *hTERT* expression. Previously, it was found that the addition of sanguinarine at 10 μM has strong affinity for human telomere repeats and *c-MYC* promoter sequence, enabling the forming of a G-quadruplex structure in vitro [36]. Isoquinoline alkaloids, represented by sanguinarine, were found to selectively recognize the telomeric G-quadruplexes in vitro and inhibit telomerase in MCF-7 cells [37]. In brief, the G-quadruplex is a common target for telomerase or other reverse transcriptases. Moreover, c-myc is an essential transcription factor in cell growth, acting via regulating the expression of related genes, including *hTERT*. The promoter region of *c-MYC* also contains abundant G-quadruplex motifs [38]. Our work suggested that SC retards HTC75 cell growth and represses the expression of hTERT and *c-MYC* (Figure 3D and Figure 4B). In contrast to the reported 10 μM concentration of sanguinarine that recognizes the G-quadruplex in vitro, the effective dosage of SC as a telomerase inhibitor in our system was lowered to 1 μM in various cells. Notably, cells were unable to survive at a dosage/concentration of over 4 μM. Nevertheless, we wondered whether telomerase could be inhibited in our system by SC via G-quadruplex binding or not.

Thus, we performed IP-TRAP and telomerase reconstitution assays. Firstly, we want to evaluate the inhibitory effects of the compound in vitro. The schematic experimental processes of IP-TRAP are depicted in Figure 5A. The immunoprecipitated telomerase complex from hTERT-overexpressed cells was incubated with serial dilutions of the compound for the telomere extension reaction. SC displayed a dose-dependent inhibitory effect on the activity of immunoprecipitated telomerase holoenzyme. The IC50 to immunoprecipitated telomerase in vitro was 1.40 μM (Figure 5A), which is close to the IC50 value when in cell culture (1.21 μM in Figure 3A). This IC50 was much lower than the concentration necessary for SC to bind the G-quadruplex.

We next purified the GST-tagged hTERT protein and in vitro transcribed hTERC, and then incubated them in a water bath. Subsequently, the addition of the compound significantly impeded the reconstituted telomerase activity in a concentration-dependent manner. Surprisingly, the suppressive effect of the compound on the reconstituted telomerase activity in vitro was exhibited at the nanomole level (Figure 5B). In conclusion, we identified that sanguinarine chloride directly inhibits telomerase at a concentration much lower than the 10 μM reported in vitro.

To investigate whether the cellular telomerase inhibition by the compound depends on telomeric G-quadruplex formation, we carried out a series of biochemical assays. Based on a previous work [36], we synthesized a human telomeric oligonucleotide (Telo24) labeled with Cy5 fluorophores. A Microscale Thermophoresis (MST) assay showed the dose-dependent binding of the compound SC to the telomeric G-quadruplex DNA. Unexpectedly, the EC50 (half maximal effective concentration) to telomeric G-quadruplexes was 100 times more than the IC50 to telomerase in cancer cells (120 μM in Figure 5C vs. 1.21 μM in Figure 3A). Thioflavin T (ThT) is a fluorescent dye used to sense G-quadruplex structures, especially in human telomeric DNA [39]. The fluorescence signal of ThT showed no difference after treatment with 1 or 2 μM of the compound or DMSO, while Pyridostatin, a G-quadruplex DNA-stabilizing agent, significantly enhanced the cellular ThT signal intensity (Figure 5D). BG4 is an antibody specific to the G-quadruplex structure. Immunofluorescence was used to visualize and quantify the cellular co-localization of G-quadruplex motifs and telomeres (indicated by an antibody against the telomeric repeat binding factor 2). Following treatment with 1 μM SC, the cell were comparable to in the DMSO control, both in the BG4 foci and in the colocalized foci of BG4 and TRF2 (Figure 5E). Furthermore, SC at high concentrations (more than 10 μM) could emit an autofluorescence signal, thus fluorescence polarization assays were carried out. The binding curve implied that SC could directly interact with TRBD of the hTERT protein in vitro (Figure 5F).

Taken together, the results showed that exposing cells to 1 μM SC does not change the formation of the G-quadruplex, and indicated that the compound at the concentration of 1 μM suppresses the telomerase activity in cancer cells by directly binding to the hTERT protein.

### 3.6. Assessment of Safety and Antitumor Efficacy of SC

As a matter of fact, sanguinarine has shown potential antitumor value in animal models [40,41]. In our system, we also needed to evaluate its safety performance and antitumor efficacy in vitro and in vivo. Firstly, we detected the cell viability of three SC-treated human primary cells with no telomerase expression. The growth of human skin fibroblasts (HFs) and HUVSMCs was analyzed via the CCK-8 assay kit, and the viability of PBMCs was traced based on CFSE labeling. All the results pointed to the safe and non-poisonous characteristics of SC in relation to primary somatic cells at a low dosage (Figure 6A,B). The cell cycle and apoptosis assays performed in the HFs suggested no increased apoptosis or cell cycle arrest was induced (Figure 6C,D). In the tested cells, 1 or 2 μM of SC had no proliferative inhibition effect.

MDA-MB-231 is a triple-negative breast cancer cell line commonly used to represent one kind of advanced breast cancer. Triple-negative breast cancer is considered to be the most dangerous because of its aggressive behavior, the lack of an effective therapy, and the high mortality in clinic. Since we found that SC works effectively in breast cancer cell lines (Figure 2F), we next wondered about the anti-tumor effect of SC. MDA-MB-231 cells were seeded into soft agar medium with drug treatment. After 14 days of culture, in the presence of 1 or 2 μM SC, a dramatic reduction in MDA-MB-231 cell-derived colonies was observed in comparison with the control (Figure 6E). Moreover, MDA-MB-231 cells were transplanted into nude mice in situ to evaluate the compound’s capability of suppressing tumorigenesis. The compound (final concentration ~1 μM in blood) was injected intravenously into the xenograft model every 3 days throughout the experimental period, and the same proportion of DMSO was used as the control. The volumes of tumors and the mouse body weight were monitored before each injection. The tumors in SC-treated mice reduced 40% versus those in control mice after 24 days of administration (Figure 6F,G), whereas the body weights remained at a constant level (Figure 6H). Taken together, we see that 1 μM SC exhibited strong antitumor efficacy both in vitro and in vivo.

## 4. Discussion

Natural products and traditional Chinese medicine have been reported to exhibit various anti-cancer capabilities. The discovery of natural compounds that inhibit telomerase can lead to advancements in tumor therapy. Here, we utilized a telomerase reporter cell line indicating the expression level of endogenous *hTERT*, which can sensitively reflect telomerase modulation under physiological conditions. After stably expressing dsRed2 as an internal control, this reporter can indicate the relative expression of *hTERT* based on the ratio of MFI. The screening approach is simple, rapid, and low-cost. Given the advantages of the reporter, we carried out a high-throughput screening of the natural product library. We succeeded in finding a few small molecules that can modulate telomerase positively (data not shown) or negatively, which are promising for use in anti-aging or anti-cancer applications.

Among the eight identified candidate inhibitors, we found that Braz has been patented as a natural telomerase inhibitor previously [31], while few related experimental studies about its suppressive effect on telomerase have been published. Here, we identified that Braz could inhibit RTA by down-regulating the *hTERT* gene and retarded the cell growth via G2/M phase arrest in cancer cells. However, only certain types of cancer cells sensitively responded to Braz. Q-TRAP data and CCK8 assay demonstrated reduced telomerase activities and cell proliferation in HTC75, HeLa and DLD1 cells upon treatment with the compound, while MDA-MB-231, Hs578t and A549 cells have no response to 8 μM Braz (Appendix A). The effective inhibition of telomerase caused by Braz may be associated with the cytotoxicity of this compound, implying that the telomerase repression by Braz is likely operated via an indirect or complex mechanism. Furthermore, telomerase activity was inhibited in cancer cells following a long-term treatment with Braz, whereas telomere shortening was not observed (Appendix A). Isolated from Caesalpinia sappan L. [42], Braz displayed antitumor abilities by inducing apoptosis and cell cycle arrest [43]. How Braz suppresses telomerase and its anti-carcinogenesis role in vivo remain to be further studied in the future.

SC was another telomerase inhibitor candidate with leading inhibition that we identified from the library. SC exhibited a broad effective range in cancer cell types and an acute inhibitory effect on telomerase and cancer cell growth. This compound down-regulated *hTERT* expression, which may be mediated by directly changing the *hTERT* transcriptional activity because of the decreased expression of several *hTERT* regulatory transcription factors, including the c-myc family, p65 et al. The oncogene *c-MYC* is a common and essential factor in the modulation of hTERT/telomerase. P65 is a subunit of the NF-κB complex and can target the remote region of the *hTERT* promoter (near −600 bp) to regulate its transcription [44]. Moreover, p65 could interact with hTERT to facilitate it transporting into the nucleus [13]. Reintroducing c-myc or p65 into the compound-treated cells indicated that the telomerase inhibition could only be rescued by p65, suggesting SC suppresses telomerase depending on p65 expression but not c-myc.

In addition, overexpressing FLAG-hTERT in the *hTERT* promoter reporter cell line caused an increase in GFP fluorescence intensity (Appendix A). This result implied that TERT transcription may enable positive feedback via its encoded protein, i.e., hTERT may act as a transcription coactivator to regulate its expression; consistently, c-myc could target the *hTERT* promoter region to activate its transcription, and hTERT could also regulate and stabilize ***c-****MYC* at the transcriptional level [11].

Apart from that, we also found that low dosages of SC could directly suppress telomerase activity both in vitro and in vivo, while not affecting telomeric G-quadruplex formation. Moreover, this inhibition in vitro was achieved at the nanomole level, much lower than the EC50 to the telomeric G-quadruplex in K^+^ solution and the IC50 to telomerase at the cellular level. Thus, SC may also directly bind to and suppress hTERT/telomerase. We performed a fluorescence polarization binding assay with TRBD and SC. The preliminary result showed a dose-dependent interaction in vitro (Figure 5F). Furthermore, we also found that this compound specifically decreases exogenous hTERT protein levels compared to the control, possibly by modulating its stability (Appendix A).

Sanguinarine has been used against different tumor or chronic diseases via different mechanisms [45]. Specifically, in breast cancer and cervical cancer, sanguinarine generates reactive oxygen species to induce apoptosis, and suppresses the NF-κB pathway to prevent metastasis [22,46,47]. In prostate cancer, the compound arrests the cell cycle by inhibiting cycle kinases and cyclins [23]. In myeloid cells, it targets the stability and phosphorylation of the IκB protein, and in certain cancers, this compound inhibits VEGF function in angiogenesis [24,48,49,50]. hTERT has also been reported to participate in the modulation of angiogenesis. Considering the complicated function of sanguinarine, researchers have designed different therapeutic approaches depending on specific cancers. For instance, low concentrations of sanguinarine could synergistically enhance the therapeutic efficacy of the chemotherapeutic agent doxorubicin in drug-resistant leukemia cells [21,51,52,53].

Different from other anti-tumorigenesis studies of sanguinarine, which were carried out by inducing apoptosis and cell cycle arrest at high concentrations [54], the administration dosage of SC was much lower in our system. Cancer cells, chronically exposed to the compound at a lower concentration for a long time, can show remarkably shortened telomeres, consequently inducing cancer cell senescence.

In conclusion, SC displayed an inhibitory effect on *hTERT* expression and telomerase activity that slowed down cell growth. Long-term treatment with SC induced changes of cell morphology and triggered senescence events, including an increase in SA-β-gal activity, the up-regulation of the expression of p16/p21/p53 pathways, progressive telomere dysfunctions (TIFs), and telomere shortening. Together, all these events triggered by SC led to senescence in cancer cells, thus blocking their progression. SC inhibits telomerase by dual functions, and its antitumor effect is potent and safe.

Previously, the sanguinarine-added mouthwash product Viadent^®^ has been reported to be associated with age-related leukoplakia, indicating its pre-neoplastic adverse effects. Oral leukoplakia is a classic symptom in dyskeratosis congenita patients [55]. Our findings show that long-term SC treatment will decrease telomere length, suggesting human adult stem cells may also be affected by long-term treatment with sanguinarine. This may explain the mechanism of the adverse effects of the Viadent^®^ mouthwash product. More precise works judging the appropriate dosage, duration and drug delivery system will improve its application in pharmacological contexts.

## 5. Conclusions

Robust telomerase activity is a common feature shared by 90% cancers. Our study identified SC at the precise dosage as an effective telomerase inhibitor, with anticancer applications. Historically, telomerase inhibitors have shown unsatisfactory performances in clinic, although they have exhibited strong suppressive effects at the cell or animal level. Therefore, as a natural telomerase inhibitor with dual functions (regulation on the mRNA and protein levels) and little proliferative inhibition effects on somatic cells, SC is potent and safe, providing a potential therapeutic approach for human malignancies. Our study proposes a prolonged treatment approach using SC to induce cancer cell senescence. Anti-tumor drugs such as SC may be synergistically used with senolytics that kill senescent cells to improve the efficacy. Additionally, the precise dosage and duration of SC application in cancer therapeutics need to be considered in future research.

## Figures and Tables

**Figure 1 cells-11-01485-f001:**
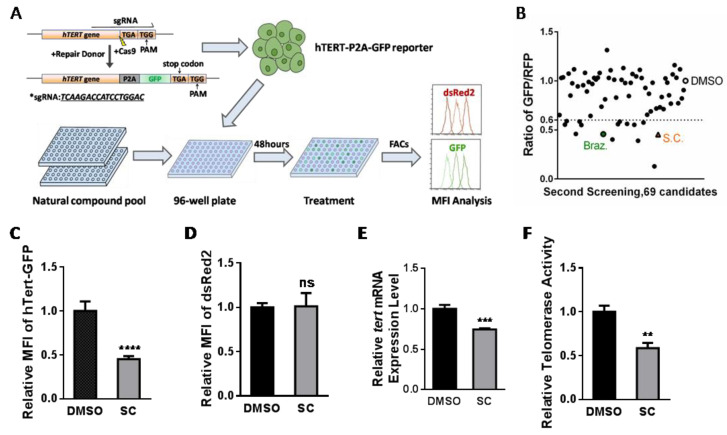
Screening of natural *hTERT* inhibitors and their verification in the endogenous hTERT-P2A-GFP knock-in HEK293Treporter cell line. (**A**) Schematics of the endogenous hTERT-P2A-GFP HEK293T reporter cell line construction and the screening strategy of a natural product pool for telomerase modulators. Two rounds of screening were carried out and compounds with GFP/RFP ratio ≤ 0.6 were selected as candidates. (**B**) Results from the second round of compound screening, with two candidates highlighted as potential inhibitors. The orange triangle indicates SC while the green rectangle indicates Braz. (**C**,**D**) Mean fluorescence intensity (MFI) quantification of endogenous hTERT-GFP (**C**) and internal reference dsRed2 (**D**) after 1 μM SC treatment for 48 h. (**E**) *hTERT* mRNA level by quantitative real-time PCR of reporter cell line upon the treatment of SC (1 μM) for 24 h. DMSO served as the control group. (**F**) Real-time quantitative telomeric repeat amplification protocol (Q-TRAP) assay in reporter cells treated with SC (1 μM). DMSO served as the control group. The screening was performed three independent times. Real-time PCR and Q-TRAP assays are from triplicate samples (* *p* < 0.05, ** *p* < 0.01, *** *p* < 0.001, **** *p* < 0.0001).

**Figure 2 cells-11-01485-f002:**
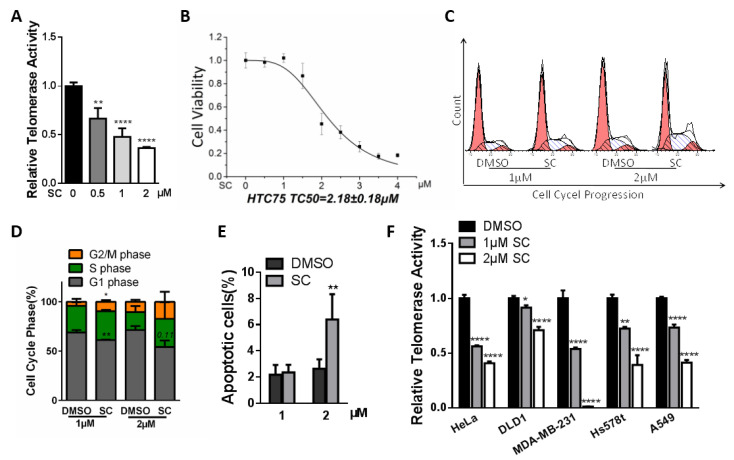
Effects of SC treatment on different cancer cells. (**A**) The inhibitory effect of SC on telomerase activity in HTC75 cells was evaluated. The relative telomerase activity (RTA) level suggested a dose-dependent suppressive effect. (**B**) The CCK-8 assay showed the cell viability curve of SC in HTC75 cells. The median toxic concentration (TC50) was 2.18 μM. (**C**) Cell cycle analysis through PI staining and the following flow cytometry for HTC75 cell treated with SC. (**D**) Quantification of cell cycle populations measured in (**C**). (**E**) Quantification of apoptotic cells (Annexin-V FITC positive cells). (**F**) The 1 μM SC treatment for 48 h hindered RTA in multiple cancer cells, normalized by RTA of the DMSO group. All the analyses were performed on triplicate samples (* *p* < 0.05, ** *p* < 0.01, **** *p* < 0.0001).

**Figure 3 cells-11-01485-f003:**
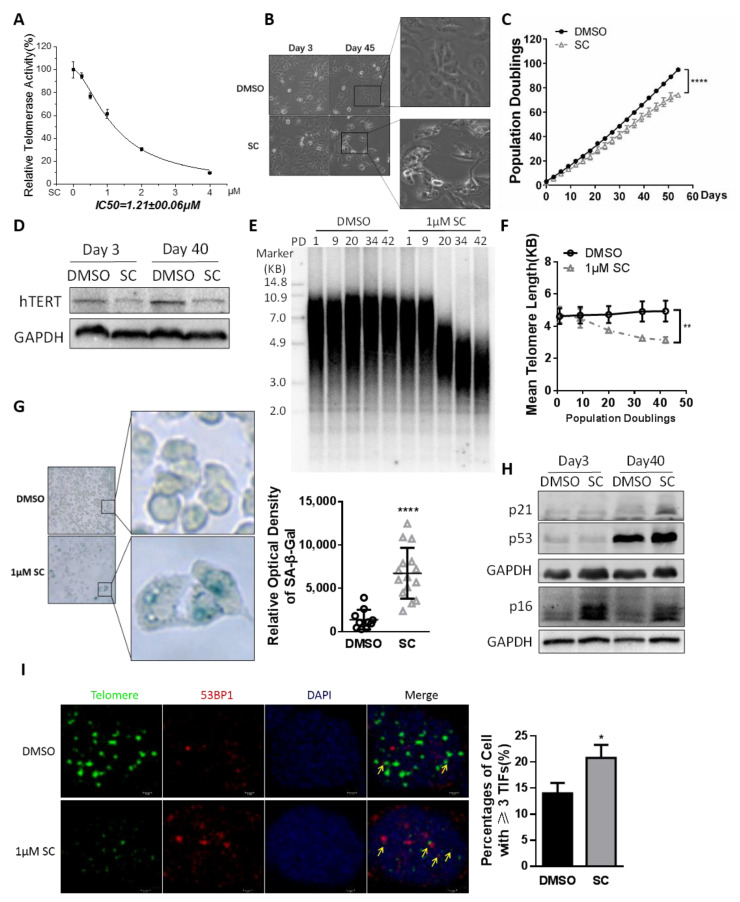
Continuous treatment with SC led to cell senescence in cancer cells. (**A**) Fitting curve of inhibitory effect of SC on RTA in HTC75 cells. IC50 = 1.21 μM. (**B**) Morphology of HTC75 cells treated with 1 μM SC versus DMSO between day 3 and day 45. (**C**) The cell growth curve of the HTC75 cells continuously treated with 1 μM SC, comparing to DMSO-treated groups. (**D**) Western blotting analysis to confirm the inhibitory effect of SC on hTERT protein level. GAPDH served as an internal reference. (**E**) HTC75 cells continuously treated with 1 μM SC were analyzed by terminal restriction fragment (TRF) assay. (**F**) Quantitative average telomere length from (**E**). (**G**) SA-β-gal staining assay to identify cell senescence in HTC75 cells continuously treated with 1 μM SC. (**H**) The senescence markers p16/p21/p53 were up-regulated in cancer cells after chronic treatment of the compound. (**I**) Telomere dysfunction-induced foci (TIFs) were analyzed using anti-53BP1 antibody (red) and PNA-conjugated telomere C strand probe (green) when HTC75 cells were treated with the compound for 40 days. Cells with TIFs ≥3 were counted for the significance test. The experiments were performed in 3 independent cell lines and the results are shown as mean ± SD, *n* = 3 (* *p* < 0.05, ** *p* < 0.01, **** *p* < 0.0001).

**Figure 4 cells-11-01485-f004:**
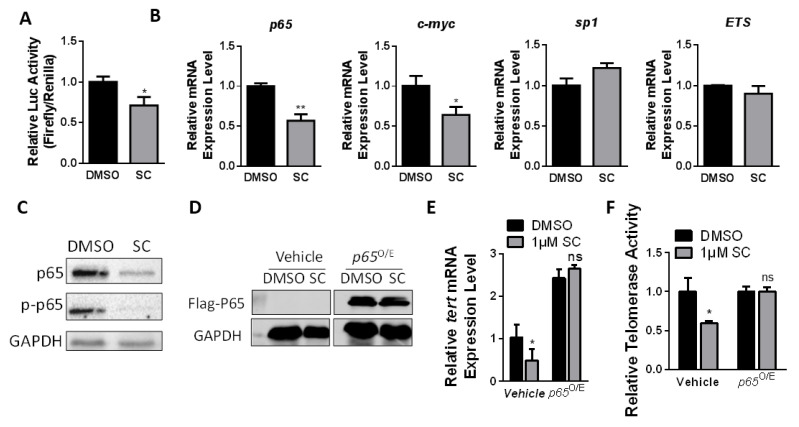
SC inhibited telomerase activity depending on *p65* expression in cancer cells. (**A**) The dual luciferase reporter assay suggested the compound has suppressive effects on *hTERT* transcriptional activities. (**B**) The relative mRNA levels of partial common transcription factors, which have been reported to regulate *hTERT* transcription. (**C**) The protein levels of pan-p65 and pi-p65 in cancer cells after 1 μM SC treatment for 48 h. DMSO served as a control. (**D**) Western blot was carried out to detect the protein level of overexpressed p65 in sanguinarine chloride-treated cells. (**E**) The mRNA level of *hTERT* was rescued in the SC-treated group by overexpressing p65. (**F**) Telomerase activity assay in p65 re-introduced HTC75 cells with SC treatment for 48 h. All the analyses were from triplicate samples (* *p* < 0.05, ** *p* < 0.01, ns means no significance).

**Figure 5 cells-11-01485-f005:**
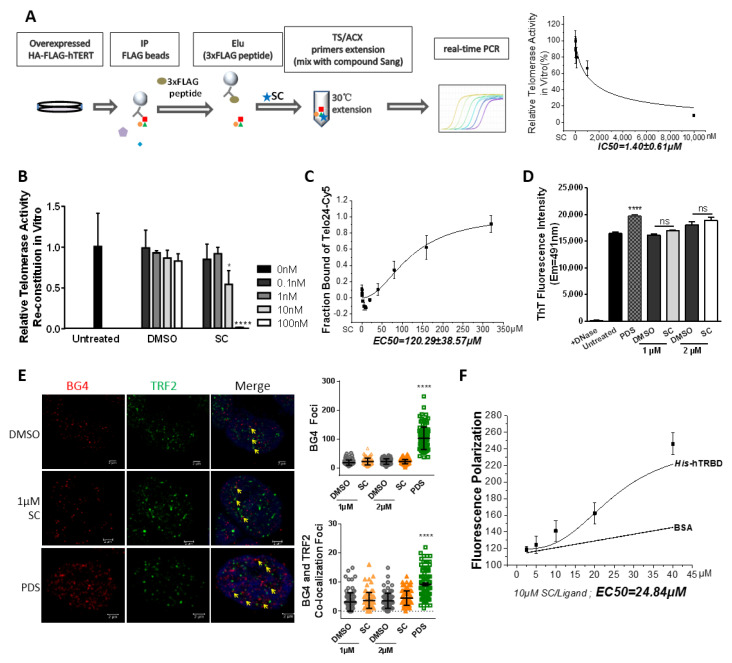
SC directly suppressed telomeric repeat extension in vitro without inducing a G-quadruplex motif. (**A**) Schematic representation of immunoprecipitation-based TRAP experiment. FLAG-hTERT-overexpressing HEK293T cells were subjected to immunoprecipitation. The elutes mixed with SC were used to perform the TRAP assay in vitro. The inhibition of natural telomerase activity by SC was presented in a dose-dependent manner, and the fitting curve showed that its IC50 to natural telomerase in vitro was 1.4 μM. (**B**) Purified GST-opTERT was incubated with in vitro-transcribed hTERC for 30 min, then mixed with SC to perform the TRAP assay. SC directly suppressed the activity of reconstituted telomerase at the nanomole level in vitro. (**C**) MST analysis of the interaction of the telomeric G-quadruplex with SC. The EC50 was 120 μM. (**D**) Detection of 2 μM Thioflavin T (ThT) fluorescence intensity at 491 nm for the whole genomic DNA in a K^+^ Tris-HCl buffer. Pyridostatin (PDS) was the positive compound used to induce G-quadruplex structure. (**E**) Representative immunofluorescence images of the G-quadruplex (recognized by BG4 antibody, red) and TRF2 (green) foci in HTC75 cells treated with SC or DMSO for 48 h. Quantification of the number of G-quadruplex foci (recognized by BG4 antibody) per nucleus in compound-treated HTC75 cells (right upper) and quantification of the number of colocalized G-quadruplex foci (recognized by BG4 antibody) and TRF2 in the nucleus (right bottom). In total, 100 nuclei were counted and statistically analyzed. (**F**) A fluorescence polarization binding assay with His-tagged hTRBD and SC was performed, and the EC50 was 24.84 μM; the BSA protein served as a negative control. All the analyses were performed on triplicate samples (* *p* < 0.05, **** *p* < 0.0001, ns means no significance).

**Figure 6 cells-11-01485-f006:**
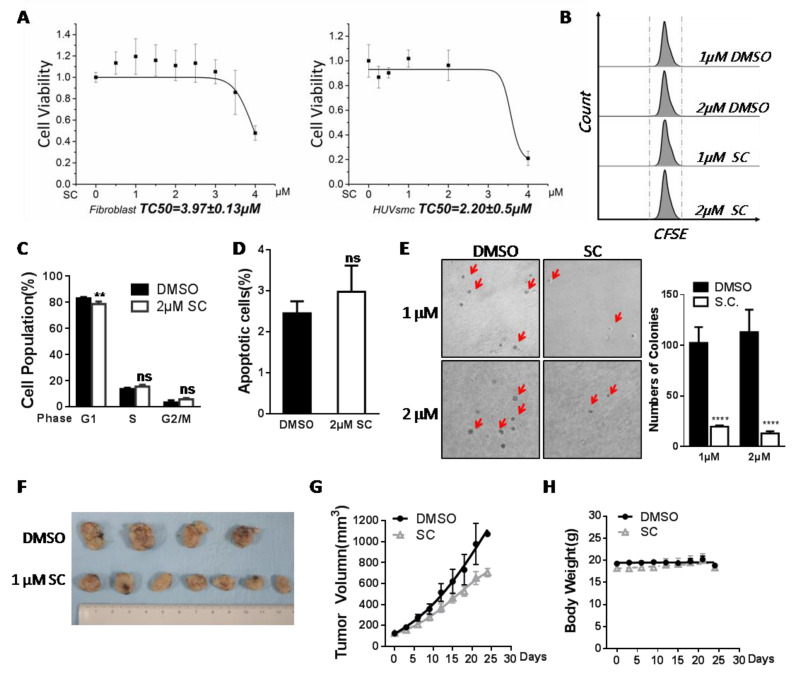
Application of SC to the inhibition of tumor formation. (**A**) the CCK-8 assay showed no proliferative inhibitory effect of SC in human primary skin fibroblast cells (**left**) and HUVSMC (**right**) at the concentration that was effective in inhibiting telomerase activity in cancer cells (data shown in Figure 3, IC50 to telomerase = 1.21 μM). (**B**) Human PBMCs were measured by CFSE labeling and then treated with 2 μM SC for 72 h. The similar characteristics of the histograms indicate that SC does not affect the proliferation of PBMCs. (**C**) Cell cycle PI staining assay was performed in fibroblast with 2 μM of compound. (**D**) Apoptosis analysis for human fibroblasts following SC treatment. (**E**) Representative photographs of the colonies from the soft agar colony formation assay in MDA-MB-231 cells treated with SC or DMSO (control). The red arrows indicate the formed cell colonies. The statistical number of colonies formed in ten randomly visual fields was quantified. (**F**,**H**) MDA-MB-231 cells were used to establish an orthotopic xenograft model in nude mice. Here, 1 μM blood concentration of SC or DMSO (vehicle) was injected into the tail vein. The total blood volume of a mouse was estimated as 7% of the body weight. A representative picture of the developed tumors of each group (**F**); tumor volume was measured every 3 days (**G**), and the body weights of xenograft nude mice was measured before every injection (**H**). All cellular assays were performed on triplicate samples, and the animal experiment was carried out in 4–7 mice (*n* = 4 in control group; *n* = 7 in SC treatment group; four sets of data were used in the volume and body weight assays, ** *p* < 0.01, **** *p* < 0.0001, ns means no significance).

## Data Availability

The data published in this study are available from the corresponding authors upon reasonable request.

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
