# Peer review of "Natural Product Library Screens Identify Sanguinarine Chloride as a Potent Inhibitor of Telomerase Expression and Activity"

_cells, 2022, doi:10.3390/cells11091485_

Round 1

Reviewer 1 Report

This is an interesting and well conducted study. The manuscript is well written, but requires a number of language improvements (singular/plural/articles, inconsistent use of times). It is a strength of the study to describe everything from the screening to cellular in vitro as well as also a xenograft mouse model. The results seem solid with good use of controls (positive as well as negative), multitude of different assays in rather good quality.

Here is a list of issues that need to e addressed:

  1. Please mention that p65 is a NFkappaB subunit in the abstract.
  2. Please use italic for gene names and "in vitro/in vivo" 
  3. "Telomeres" are plural-there are 92 per normal cell!
  4. line 54/55: Although your sentence is relevant for telomere inhibition in cancer cells, scientifically, telomere shortening occurs due to the end replication problem (ERP) in ALL dividing cells, also those that never had telomerase activity (TA). Lack or decrease of TA just prevents their re-elongation after DNA replication. Please correct and rephrase this statement.
  5. Imetelstat binds with high affinity to hTERC and is a competetive inhibitor of telomerase activity (TA). Please cite one of the original papers. Possibly it might thereby indirectly/secondarily also telomerase assembly, but this is not its primary target.
  6. line 218: What exactly do you mean with "monoclonal cell line"? What cell line does it refer to? In general, due to their high genomic instability the majority of cancer cell lines is aneuploid and changes continuously in culture. Did you perform a karyotype analysis at different time points? Please elaborate and give more details and a rational why this is important for your experiments or remove the term.
  7. It is important to state  numbers of experiments and whether these are technical or biological repeats for all experiments in the figure legends.
  8. Please indicate in the figure legend what cell line(s) have been used in fig. 1.
  9. line 264: In my view a "toxic effect" is rather unspecific. Thus, the term "proliferative toxic effect" seems not indicated.
  10. In fig. 3B hardly any differences between cells (treated/untreated) are visible. Thus, it seems problematic to say that SC-treated cells presented senescence-like characteristics or at least give more details-probably you refer to an enlarged, flattened morphology? It is hard to see on the phase contrast images.
  11. line 309: It is not correct that telomere shortening (TS) induces genomic instability (GI) since senescence is there to prevent GI. TS induces a DNA damage response (DDR) that, when the DNA damage cannot be repaired, results in an irreversible growth arrest=senescence. Please correct the statement.
  12. Please be aware that only genes are expressed, thus for proteins or MRNA do not use expression but "levels or amounts". For various graph axis use either "gene expression" or "mRNA level". Likewise for the figure legend 3 C where you describe TERT protein, NOT its gene expression.
  13. The description of Western blotting is missing from the methods section. Importantly, please describe what anti-TERT antibody you have used and how did you validate its specificity.
  14. line 392: I don't think it is possible to IP an enzymatic (telomerase) activity. Please rephrase and correct. You probably mean the hTERT protein?
  15. For your figure legend 5 the heading is a bit confusing as TA in most functions including the enzymatic has absolutely nothing to do with telomeric G-quadruplexes which just block access of telomerase to telomeres. Please correct and rephrase. You previously had already shown that SC also decreases hTERT expression which also is independent of any telomeres or G-quadruplexes.
  16. line 462: "we wondered ABOUT the...effect"
  17. line 487: What does mean "tumor volume EXO"?
  18. line 496: I think that the MFI RATIO is just there for normalisation while hTERT expression is measured as the MFI.
  19. line 534: Please replace "accidentally" with perhaps "incidentally"?
  20. line 544: "Sanguinarine has been USED against different tumor..." You cannot "report anything against tumors". 
  21. line550: What exactly do you mean with "Paranthetically"? I don't think it is the correct term as it means "in brackets".
  22. line 552: Please replace "cases" with "approaches"
  23. line 559: Better say "shorten" than "attrite"
  24. line 566: Better use "progression" than "developemnt" for tumors
  25. line 574: do you really mean "agricultural" use? You did not talk about farming in this study. Please correct.
  26. In the discussion you missed to discuss the important fact that your experiments showed that at least in vitro there don't seem to be any adverse effects on normal somatic cells. This is very important and remarkable since various other telomerase inhibitors (TI) including imetelstat showed such effects in clinical trials. You should also detail this in your conclusion  where you talk about "unsatisfactory performance" of other TI.
  27. line 585: Please replace "test" with "research"
  28. In general, your English language and grammar requires extensive improvements and needs to be corrected by a native speaker as a frequently clumsy and incorrect language contradicts your good quality scientific content. In particular in the discussion, often even words are written together that need to be apart, your use of articles, mix of times (present and past mixed) and singular/plural need to be substantially improved.
  29.  

Reviewer 2 Report

This report describes the screening platform and the characterization of two natural product telomerase inhibitors in human cancer cell models. The overall conclusion is that sanguinarine chloride (SC), a compound found in some mouthwash formulations and studied for its cytotoxic effects on various cancer types, can repress TERT transcription and directly inhibit telomerase activities at concentrations that are much lower than previously reported for its cytotoxic effects. 

The screening was performed in HEK293 cells with CRISPR-knock-in GFP following the coding sequence for TERT, with GFP fluorescence reporting the mRNA expression of TERT in its endogenous locus. From a library of 800 commercially available natural compounds, 69 exhibited a minimal 40% decline in GFP signals upon treatments and 8 of them passed reproducibility experiments. 

Characterization of short-term effects of these telomerase inhibitors was conducted in Figure 2. Cytotoxicity was induced with short-term (48h) treatment in HTC75 cells with an LD50 of 2.18uM. Cell-cycle analyses revealed accumulations of arrested cells @ G2/M. 48h treatment with SC demonstrated a dose-response inhibition of TRAP in HTC75 and HeLa, DLD1, MDA-MB-231, Hs578t and A549 cells, demonstrating that SC’a telomerase inhibition covers more cancer models than that displayed by BRAZ. Longer-term effects of SC exposure were shown in Figure 3. Treatment of HTC75 cells with 1uM Sc for 60days in culture resulted in morphology changes similar to the induction of proliferation senescence, which was confirmed with telomere attrition, the expression of the senescent marker, p53 activation and p21 expression, as well as the senescence-associated beta-galactosidase activity. Specifically, telomere-dysfunction foci were also up-regulated in treated cells.

To map the mechanism of TERT transcriptional regulation by SC, the expression levels of four TERT transcriptional factors were assessed, with both c-myc and p65 showing SC-dependent reduction in steady-state mRNA (c-myc and p65) and protein (p65) levels. Over-expression of a recombinant p65 rescued SC-treatment induced reduction in telomerase activity, suggesting the loss of p65-dependent transcriptional activation of TERT is responsible for this effect (Figure 4). Figure 5 evaluates whether the G-quadruplex-stabilizing properties of SC could explain its inhibitory effects against TRAP activity. The authors concluded that SC stabilized G-quadruplex at a concentration much higher than that used in telomerase inhibition and TIF formation, thus unlikely to be the mechanism of action.

Finally, to demonstrate the therapeutic potential of SC, the authors showed that SC exposure to human primary fibroblasts and PBMC did not induce overt toxicities. At the same time, the growth of 3D cell culture (spheroid/agar colony formation) and mouse xenograft models of the TNBC line MDA-MB-231 were inhibited significantly (Figure 6). 

The Study is fairly straightforward, and the data presentation is clear. All biochemical methods for cytotoxicity, apoptosis and cell-cycle profile measurements were adopted from classical studies or commercial kits. As these methodologies have been used for decades, there are no concerns about assay development. The authors convincingly showed that treatment with SC reduced the viability of HTC75 and MDA-MB-231 cells in vitro and with xenograft animals and demonstrated that this compound inhibits telomerase expression and activity.

This reviewer found the following assertions need to be clarified for the general readership: 1) SC may have additional biological effects not revealed by the current study, and 2) the mechanism by which SC inhibits TERT (and p65/c-myc) transcription was not described. These specific points are discussed below:

Major:

1) the current Study does not comprehensively reveal SC biological effects

Past studies have demonstrated that cellular effects of telomerase inhibitors, such as BIBR1532 and GRN163L, appear only after a sufficient lag-time for telomere attrition. Thus it is surprising to see the rapid (48h) induction of cytotoxicity and cell cycle arrest in HTC75 fibrosarcoma cells receiving SC alone (Figure 2). However, this rapid onset of treatment effects can be explained by the additional data on SC’s effects on global transcriptional regulation (through TERT or otherwise). Treatment with SC induced transcriptional silencing of p65 and c-myc, which are transcriptional factors frequently implicated in carcinogenesis. Regardless of whether down-regulation of c-myc and p65 are direct functions of SC or through the positive feedback of inhibiting TERT’s non-canonical transcriptional activities, reductions of these TF’s steady-state levels are likely anti-proliferation in rapidly dividing cells. Accordingly, exposure to SC will impact transformed as well as normal cells through multiple mechanisms in addition to telomerase inhibition. In lieu of a genomic study on SC’s effect on the transcriptome, these complications warrant further discussions and a substantial clarification that SC is not solely a telomerase inhibitor.

2) There are remaining uncertainties in the mechanism of SCs transcriptional regulation activities

The authors demonstrated that recombinant expression of p65 in SC-treated cells can restore TERT expression levels. This mechanism is further explained by a possible positive feedback system of TERT's potential non-canonical activity in the transcriptional regulation of the expression of p65 and c-myc. Even if TERT may have transcriptional regulatory effects on myc and p65 expression, it remains unresolved how SC treatment may affect these regulations. The authors demonstrated that the SC dosage used for the inhibitory studies is insufficient to induce G-quadruplex formation, thus ruling out SC’s effects on transcriptional inhibition through GQ formation at the respective promoter sites. The authors should discuss how SC could have inhibited transcription of these factors to strengthen their observations and conclusion.

Minor:

  1. SC and Braz were selected based on their “outstanding inhibitory effects” on TERT expression; however, from the data shown in Table S1, it is unclear why these two compounds are more favorable than the others, as the GFP/dsRED2 ratios for these two compounds are less than that for daunorubicin, and similar to peruvoside (Figure 1, S1 and Table S1)
  2. Apoptosis induction and cell-cycle arrest profiles were demonstrated using a single cancer model, HTC75. Since there are questions on the lack of a lag-time associated with telomerase inhibition and SC effects, it may be complementary to show the induction of apoptosis and cell-cycle arrest at the G2/M-phase in other cancer cell lines treated with the compound.

Reviewer 3 Report

The manuscript by Yan et al., describes a hTERT promoter screen where they identify sanguinarine chloride (SC) as a regulator of hTERT expression and telomerase activity. Their results are compelling and of significant interest to the field of telomere biology. I only have a few minor comments I'd like addressing.

1) The English used throughout the manuscript need significant editing. There are numerous typos and spelling mistakes. This is very evident in the intro and discussion sections.

2) Figure 1G is missing. 

3) Can SC suppress hTERT expression in a dose dependent manner to match the reduced telomerase activity seen in Fig 2A?

4) Does telomere length "stabilize" or do the telomeres continue to shorten indefinitely with SC treatment (Fig 3E)? I would think that they would stabilize as you still see telomerase activity.

5) Can you perform hTERC dot blots (or similar) on IP telomerase samples from cells exposed to SC and look for changes in the ratio of hTERC:hTERT? This would support the results that suggest SC impairs telomerase assembly.

Reviewer 4 Report

The authors examine inhibitory effects and mechanisms of two previously identified telomerase inhibitors (in natural product library screening). They perform a comprehensive analysis of symptoms induced by these compounds (e.g., telomere length, telomerase activity, hTERT expression, senescence symptoms...). I find the study useful and informative, technically sound and worth publishing after the following revisions:

  1. The Title of the paper is misleading – it suggests the paper describes identification of telomerase inhibitors in natural product library screens. However, it rather describes a closer characterisation of two previously identified candidates. Please, change the title so that it better corresponds to the MS.
  2. Methods – TRF analysis – evaluation of the mean telomere lengths may be biased due to well known reasons (e.g., the shorter telomeres also provide weaker signals and are thus underrepresented...). Therefore, I suggest to use the online tool WALTER (Lycka et al., BMC Bioinformatics, 2021) to re-evaluate the result shown in Figure 3F.
  3. Discussion-Conclusions: I suggest to present main conclusions on mechanisms/steps in telomerase regulation where the examined inhibitors act, as a Figure - to make the take home message for the readers even more clear.

minor - language revisions and correction of typos needed throughout the MS, e.g.:

p.2: is chemically modified oligonucleotides

L442: “...the results showed that cells exposing to 1 μM SC does not change 442 the formation of the G-quandruplex structure...“

L569: “dyskeratosiscongenita“

Round 2

Reviewer 1 Report

The authors have addressed most of my comments. but not all. The sentence in lines 59-61 has the same content and is even worse than the previous version as telomerase activity (TA) but NOT hTERT expression is responsible for telomere maintenance. HTERT can have non-canonical, not telomere-related functions and thus is not a synonymous for TA. Moreover, , in connection with other terms such as "shortening" etc, it should be "telomere shortening/extension". You better revert back to the initial phrase as my comment related to the fact mentioned by you at the beginning of the introduction that the molecular cause of telomere shortening (TS) is the end replication problem (ERP) which is independent of the presence or absence of TA while the latter just counteracts TS when present.

Furthermore, in your response to comment 10 regarding cell morphology you describe that as "shrinking and irregular" which is NOT a feature of senescence where cells get larger and have a more flattened morphology, but rather suggest cell death and toxicity. Thus, please remove any mentioning of senescence and describe your changes in morphology as they are, but not as anything senescence-related.

Unfortunately, by stating in the original manuscript (sorry, i had missed that) AND in the response letter that you had used just 3 technical repeats for your experiments you then just changed "technical" to "biological" repeats. This is IMPOSSIBLE and seems like a very serious scientific misconduct bordering on fraud. Please honestly state that you had just used 3 technical repeats and discuss this as a serious scientific limitation in the discussion as for good science you have to perform. independent biological repeats.  In general, such a scientific misconduct can result in your study and manuscript to be rejected. It is also a misregard of the intelligence of the reviewer to assume that they will not spot such an impossible change of the complete design of your study which cannot be amended retrospectively and you had confirmed the use of technical repeats in your response letter. Such a behaviour also casts doubts about your scientific trustworthiness and this is a pity since you presented an otherwise very nice study with interesting data.
